# Cyanobacterial Harmful Bloom Lipopolysaccharides Induce Pro-Inflammatory Effects in Immune and Intestinal Epithelial Cells *In Vitro*

**DOI:** 10.3390/toxins15030169

**Published:** 2023-02-21

**Authors:** Veronika Skočková, Ondřej Vašíček, Eliška Sychrová, Iva Sovadinová, Pavel Babica, Lenka Šindlerová

**Affiliations:** 1Department of Biophysics of Immune System, Institute of Biophysics of the Czech Academy of Sciences, 61200 Brno, Czech Republic; 2Department of Experimental Biology, Faculty of Science, Masaryk University, 62500 Brno, Czech Republic; 3RECETOX, Faculty of Science, Masaryk University, 62500 Brno, Czech Republic; 4Department of Experimental Phycology and Ecotoxicology, Institute of Botany, 60200 Brno, Czech Republic

**Keywords:** cyanobacteria, cyanobacterial harmful bloom, lipopolysaccharide, inflammation, intestine, macrophage

## Abstract

Freshwater cyanobacterial harmful blooms (CyanoHABs) produce a variety of toxic and bioactive compounds including lipopolysaccharides (LPSs). The gastrointestinal tract can be exposed to them via contaminated water even during recreational activities. However, there is no evidence of an effect of CyanoHAB LPSs on intestinal cells. We isolated LPSs of four CyanoHABs dominated by different cyanobacterial species and LPSs of four laboratory cultures representing the respective dominant cyanobacterial genera. Two intestinal and one macrophage cell lines were used to detect *in vitro* pro-inflammatory activity of the LPS. All LPSs isolated from CyanoHABs and laboratory cultures induced cytokines production in at least one *in vitro* model, except for LPSs from the *Microcystis* PCC7806 culture. LPSs isolated from cyanobacteria showed unique migration patterns in SDS-PAGE that were qualitatively distinct from those of endotoxins from Gram-negative bacteria. There was no clear relationship between the biological activity of the LPS and the share of genomic DNA of Gram-negative bacteria in the respective biomass. Thus, the total share of Gram-negative bacteria, or the presence of *Escherichia coli*-like LPSs, did not explain the observed pro-inflammatory activities. The pro-inflammatory properties of environmental mixtures of LPSs from CyanoHABs indicate their human health hazards, and further attention should be given to their assessment and monitoring.

## 1. Introduction

Cyanobacterial harmful blooms (CyanoHABs) in freshwater reservoirs represent a serious environmental issue worldwide. They can be formed by various cyanobacterial species. The predominant cyanobacterial genera in CyanoHABs are *Microcystis*, *Aphanizomenon*, *Dolichospermum*, *Cylindrospermopsis,* and *Planktothrix* [1,2,3]. Cyanobacteria are able to produce a wide range of toxins, mostly secondary metabolites. Some of these cyanobacterial toxins (cyanotoxins) are well described and characterized (e.g., microcystin or cylindrospermopsin), while others have been discovered relatively recently with limited information thus far regarding their environmental occurrence, fate, and health effects and risks (e.g., puwainaphycins and minutissamides) [2,4,5]. In addition to secondary metabolites, CyanoHABs are also a source of endotoxins, lipopolysaccharides (LPSs). LPSs are an integral part of the outer membrane of the cell wall of both cyanobacteria and heterotrophic Gram-negative (G-) bacteria [6,7]. Therefore, LPSs are inherently present in any locality with CyanoHAB occurrence [6,7]. In addition to cyanobacterial LPSs, heterotrophic G- bacteria are naturally associated with CyanoHABs and interact with cyanobacteria [8,9,10], representing another source of LPSs during CyanoHAB events. In some instances, pathogenic G- bacterial species, e.g., *Acinetobacter*, *Pseudomonas*, *Aeromonas*, or *Vibrio*, were detected in CyanoHABs [11]. LPSs from both cyanobacteria and G- bacteria consist of the same general structural regions, i.e., O-specific antigen, core oligosaccharide (outer and inner core), and lipid A. However, structures of a few cyanobacterial LPSs that were characterized thus far revealed various specific features that are not typical for G- bacteria [6,12,13,14].

LPSs of some G- bacteria are known to be ligands of Toll-like receptors (TLRs), especially TLR4 [15,16]. TLRs are able to detect various molecules present in bacterial cell walls, membranes, or flagella. TLR4 is present on plasmatic membranes of various cell types, mainly on immune and epithelial cells including macrophages and enterocytes [17]. Enterocytes are the most numerous cell type in the intestinal epithelial monolayer, making a physiological barrier between the outer and inner environment of the organism. When activated by TLRs, enterocytes produce, among other cytokines, chemokine interleukin (IL) 8. It is important for neutrophil recruitment, which can result in inflammation. In close proximity to the enterocytes, tissue macrophages are present [18]. Once enterocytes are activated, activation of these macrophages occurs and they produce other important pro-inflammatory compounds such as tumor necrosis factor alpha (TNFα), IL-6, or nitric oxide (NO) [19]. It is well documented that even short recreational exposure to cyanobacterial water blooms during swimming [20,21,22], water-skiing [22], or canoeing [23] is associated with gastroenteritis [2]. Although it has been hypothesized that these effects can be connected to CyanoHAB LPSs, surprisingly, there is little experimental evidence to support that complex CyanoHAB LPSs or cyanobacterial LPSs, a potential TLR activator, can affect the intestinal cells and possibly cause gastroenteritis and other health problems [14]. 

On the other hand, over recent years, several *in vitro* studies have confirmed the pro-inflammatory effects of CyanoHAB or cyanobacterial LPSs on other cell types. The main interest has focused on immune cells such as monocytes, neutrophils, and B cells. The activation of immune cells was observed after exposure to LPS isolated, e.g., from *Microcystis aeruginosa* [24] or *Geitlerinema* sp. [25,26] cultures as well as from environmental CyanoHAB samples [1,24]. Besides immune cells, LPSs isolated from *Geitlerinema* sp. or natural CyanoHABs were able to induce a pro-inflammatory response in airway epithelial cells *in vitro* [27,28], and *Geitlerinema* sp. LPSs also caused lung inflammation in a mouse model *in vivo* [28]. However, we are not aware of any studies dealing with the effect of the cyanobacterial LPS on intestinal epithelial cells.

In this work, we aimed to describe the pro-inflammatory *in vitro* effects of CyanoHAB LPSs on enterocytes and macrophages. The environmental mixtures of LPSs were isolated from CyanoHABs collected in recreational freshwater reservoirs in the Czech Republic. Moreover, we studied the pro-inflammatory potential of LPSs from laboratory cultures of four cyanobacterial strains representing genera dominating the environmental samples, i.e., *Microcystis*, *Aphanizomenon*, *Dolichospermum*, and *Planktothrix*.

## 2. Results

### 2.1. LPS Samples Characterization and Their Endotoxin Activity

The CyanoHAB LPS was isolated from four environmental water bloom biomass samples from various localities in the Czech Republic. Each sample was dominated by different cyanobacterial species described in detail in Table 1. The amount of isolated LPS varied from 7 to 16 mg LPS/g of the biomass dry weight (Table 1). Moreover, LPSs from cyanobacterial laboratory cultures of respective genera dominating the studied CyanoHABs were isolated (Table 1). Commercially available LPSs from G- bacteria *Escherichia coli* (EC) and *Pseudomonas aeruginosa* (PA) were used as positive controls.

A PyroGene^TM^ assay was performed and the endotoxin activity was calculated to describe the pyrogenicity of the LPS. Among CyanoHABs, the highest endotoxin activity was manifested in the case of HAB-Mi, being ten times higher than the activity of LPS EC and almost one hundred times higher than that of LPS PA. On the contrary, the other three LPSs showed very low or almost no endotoxin activity (Table 1). Among laboratory cultures, *Aphanizomenon* (Ap) and *Microcystis* (Mi) LPSs showed similar endotoxin activity, which was approximately three times higher than the activity of LPS PA and three times lower than the activity of LPS EC (Table 1). In contrast, *Planktothrix* (Pl) and *Dolichospermum* (Do) LPSs showed minimal endotoxin activity.

The cytotoxicity of all LPSs was tested by LDH assay in all *in vitro* cell models (Caco-2, HT-29, RAW 264.7) (Appendix A). Only non-cytotoxic LPS concentrations were used for further analyses.

### 2.2. Pro-Inflammatory Activation of Intestinal Cells 

Differentiated Caco-2 cells as well as confluent HT-29 cells were exposed to a range of concentrations of the LPSs and the production of pro-inflammatory IL-8 was measured by ELISA. Interestingly, not only highly pyrogenic LPS HAB-Mi but also LPS HAB-Do (in both cell lines) and LPS HAB-Pl (in HT-29 cells) were able to induce IL-8 production (Figure 1). Moreover, the effect of both active CyanoHAB LPSs was much higher than the effect of LPS EC in Caco-2 cells (Figure 1A). Although LPSs HAB-Pl and HAB-Do had dramatically lower endotoxin activity than LPS EC and HAB-Mi (Table 1), their effect on IL-8 production was comparable to or even slightly higher than those of LPSs EC and HAB-Mi in HT-29 cells (Figure 1B). The only LPS without any significant effect on IL-8 release in both cell lines was LPS HAB-Ap. Surprisingly, both cell lines reacted differently to bacterial LPSs. Caco-2 cells were much more sensitive to LPS PA than to LPS EC, in contrast to HT-29, which did not respond to LPS PA at all.

Experiments with the LPS from cyanobacteria cultures were carried out in a similar way as for CyanoHAB LPS. We observed a significantly increased level of IL-8 in medium after the exposure of Caco-2 cells only to *Dolichospermum* LPSs (Figure 2A). Similarly to CyanoHAB LPS experiments, HT-29 cells were more sensitive to cyanobacterial LPSs than Caco-2, but they were activated by *Dolichospermum* as well as *Aphanizomenon* LPSs (Figure 2B).

Furthermore, a cytokine array was performed to analyze a wider spectrum of pro- and anti-inflammatory cytokines and chemokines produced in the response of Caco-2 cells to significantly active CyanoHAB LPSs, i.e., LPS HAB-Do and HAB-Mi (100 µg/mL). The most important changes in the production of various molecules caused by CyanoHAB LPS exposures are shown in Figure 3. The whole spectrum of all compounds detected by the array is shown in Appendix A. The results confirmed increased IL-8 production induced by both CyanoHAB LPSs (Figure 3C). In addition to IL-8, a wide range of other chemokines (GROα, I-TAC, MCP-1, MIP-1α, MIP-1β, RANTES, and SDF-1) showed increased levels (Figure 3C). Similarly, both LPS samples activated the secretion of pro-inflammatory cytokines IL-1α, 1β, 5, 6, 17, 32α, and TNFα (Figure 3A). Moreover, expression of anti-inflammatory cytokines IL-2, 10, 12, 27, and sTREM-1 was induced (Figure 3B).

### 2.3. Pro-Inflammatory Effects on Macrophages

Concentrations of pro-inflammatory cytokines IL-6 and TNFα, as well as nitrite levels in the culture medium, were determined as markers of macrophage activation. We observed a significant increase in TNFα production again not only for the highest concentration of LPS HAB-Mi with high endotoxin activity but also for the highest concentration of LPSs HAB-Ap and HAB-Do (Figure 4A). Nevertheless, these effects were minor compared to effects of LPSs EC and PA. On the contrary, no elevation in IL-6 was detected after CyanoHAB LPS treatment (Figure 4B). Similarly, no studied CyanoHAB LPS was able to induce production of NO by the macrophages (Figure 4C).

In the case of cell cultures, TNFα production was induced by *Aphanizomenon* and *Dolichospermum* LPSs, and in the highest concentration also by *Planktothrix* LPS. The only inactive LPS was from *Microcystis* (Figure 4D). In contrast to water bloom LPSs, the highest concentrations of *Dolichospermum* and *Aphanizomenon* LPSs were able to also induce IL-6 production (Figure 4E). However, this increase in TNFα and IL-6 production was not associated with induced NO production that is typically observed for bacterial LPSs (Figure 4F). 

### 2.4. Biomass DNA Composition

To estimate the relative share of total heterotrophic bacteria, G- bacteria, and cyanobacteria in the water bloom biomasses used for LPS isolation, qPCR analyses using specific TaqMan probes were performed. The bacterial presence in *Planktothrix*-dominated water blooms was almost undetectable with total heterotrophic bacteria representing 1% of the quantified DNA sequences, out of which approx. 0.5% corresponded to G- bacteria (Figure 5A). The highest proportion of G- bacteria was observed in the biomass of HAB-Do, representing 50% of the quantified sequences of DNA.

We also analyzed the share of total heterotrophic bacteria and G- bacteria in the DNA isolated from cultured cyanobacterial biomasses. Interestingly, the culture of *Dolichospermum* CCALA007 providing the most bioactive LPS was also the biomass with the lowest share of bacterial DNA sequences (~1.7%) belonging primarily to G- bacteria (Figure 5B). In addition, the harvested *Microcystis* biomass contained a relatively low share of bacterial DNA (3% of total quantified DNA sequences), with G- bacteria representing 1.3%. On the other hand, the *Planktothrix* culture had the highest bacterial presence (65.41%) including G- (10.19%). In the *Aphanizomenon* culture, the bacterial DNA corresponded to one-quarter of the total DNA sequences quantified (with ~8% of G- bacteria).

### 2.5. SDS-PAGE Migration Patterns of Studied LPS 

All studied samples of LPS were separated by SDS-PAGE and visualized using Pro-Q^TM^ Emerald 300 glycoprotein staining (Figure 6). The lack of protein contamination of the samples was verified by Coomasie blue staining (Appendix A). SDS-PAGE separation of commercially available LPSs from G- bacteria revealed the presence of typical ladder-like patterns as reported previously for LPSs purified from *E. coli* [33,34] or *P. aeruginosa* [35,36,37,38]. These repetitive bands represent different types of LPS structures with a varying number of repeating sugar units in the O-specific antigen chain (i.e., smooth LPS types), as well as LPSs with only short O-antigen chains (i.e., semi-rough LPS) and lower-molecular-weight bands of LPSs formed only by the core oligosaccharide and lipid A (i.e., rough LPS type). LPSs isolated from CyanoHABs or cultured cyanobacterial strains showed sample-specific migration patterns, which were clearly different from those of *E. coli* or *P. aeruginosa* LPSs. LPSs isolated from cyanobacterial biomasses revealed primarily higher-molecular-weight bands corresponding to the smooth LPS types, ranging from LPSs with intermediate to very long O-specific antigen chains. Bands with lower molecular weights corresponding to rough or semi-rough LPS types were observed only in some cyanobacterial LPS samples (e.g., cultures of *Aphanizomenon* PCC7805 and *Dolichospermum* CCALA007), but these bands were usually less prominent than other major bands detected in the higher-molecular-weight regions. LPS HAB-Ap showed a variety of bands in the smooth LPS type region, with three intense bands occurring in the area of the intermediate O-antigen. Interestingly, a major band in the same region was also observed in LPSs isolated from the *Aphanizomenon* culture. In contrast, both LPS HAB-Pl and LPSs from the *Planktothrix* culture showed an absence of bands below a region corresponding to a very long O-antigen. LPSs of *Dolichospermum*-dominated HAB-Do showed bands above the intermediate O-antigen region. However, the culture of *Dolichospermum* CCALA007 revealed a very different LPS migration pattern, with a ladder-like structure starting from the lowest molecular weight up to a bright thick band in the long O-antigen region. LPSs from CyanoHAB and cultured biomass dominated by *Microcystis aeruginosa* had similar ladder-like migration patterns starting from an intermediate area and displaying 2–3 intense bands in the intermediate and very long O-antigen regions.

## 3. Discussion

This study aimed to examine the pro-inflammatory effects of water bloom lipopolysaccharides (LPSs) on enterocytes and immune cells *in vitro*. Accidental consumption of contaminated water during recreational activities in CyanoHAB-affected reservoirs may expose, in a mixture of other bioactive molecules and/or cyanotoxins, enterocytes to LPSs from CyanoHABs. The LPSs could activate or contribute to activation of the enterocytes and to disruptions to the intestinal epithelium [2]. An impaired intestinal epithelium can result in increased paracellular transport of LPSs, exposing immune cells residing in the near proximity [40]. LPSs were isolated from water blooms dominated by *M. aeruginosa*, *A. klebahnii*, *P. agardhii*, and *D. curvum*, common bloom-forming cyanobacterial species [4].

LPS is known as an endotoxin and its presence and activity are usually evaluated by Limulus amoebocyte lysate (LAL) assay or PyroGene™rFC Assay. For the CyanoHAB LPS, PyroGene™ was found to be a more robust and specific assay than LAL [1]. Interestingly, the activity of our LPS sample of *Microcystis*-dominated water blooms was almost two hundred times higher than the maximal endotoxin activity found among 11 CyanoHAB LPS samples isolated from *Microcystis*-dominated water blooms (range 9–190 × 10^3^ EU/mg, average 59 × 10^3^ EU/mg) [1]. On the other hand, LPSs from water blooms dominated by other cyanobacterial species (*Aphanizomenon*, *Planktothrix*, *Dolichospermum*/*Anabaena*) had endotoxin activities not only lower than LPSs from *Microcystis*-dominated water blooms [1,24], but also lower by two or three orders of magnitude than LPSs isolated from the samples dominated by corresponding cyanobacterial genera (35–270 × 10^3^ EU/mg) previously published by Bernardová et al. [41]. However, these earlier measurements were performed by an LAL assay [41] that found for LPSs from *Microcystis* on average 13-times-higher endotoxin activities than those of the PyroGene™ assay [1], and this phenomenon could possibly occur also to LPSs isolated from other cyanobacterial species. However, the differences among endotoxin activities between different samples and studies could be most likely attributed to different taxonomical compositions of respective water blooms, including cyanobacteria as well as associated G- bacteria, which show qualitative and quantitative variations depending on the season, environmental factors, cyanobacterial species, or oligotype [8,9,10,42]. Different cyanobacterial as well as G- bacterial species, strains, genotypes, or serotypes can exhibit major differences in their LPS structures [6,12,13], and also their endotoxin activities can differ by several orders of magnitude [1,41,43]. This is supported also by our study, showing major differences in the endotoxin activities of LPSs of different G- bacteria, LPSs isolated from different laboratory cultures of cyanobacteria, and different CyanoHAB LPSs. 

Furthermore, structural and/or chemical variations among the LPSs used in this study were indicated by SDS-PAGE with sample-specific LPS migration patterns and different dominant bands. In addition, previously published studies showed different electrophoretic profiles of LPSs isolated from different cyanobacterial strains. LPS core structures as well as bands corresponding to LPSs with the O-specific antigen were detected in LPSs isolated from *Synechococcus* CC9311 and WH8102 [44]. In our study, low-molecular-weight bands presumably corresponding to the core structures and short O-specific antigen chains of rough or semi-rough LPS types [33,34,35,36,37,39] were observed only in LPSs of *Aphanizomenon* PCC7805 and *Dolichospermum* CCALA007 cultures. In contrast, such low-molecular-weight bands were not found in LPSs HAB-Ap and HAB-Do, which were dominated by the smooth LPS type. However, CyanoHAB LPSs showed some similarities with migration patterns from the cultures of corresponding dominant genera, such as an intense band in the intermediate O-antigen region of *Aphanizomenon* LPS, a laddering pattern of *Microcystis* LPS starting from the intermediate region and ending up with an intense band in the very long O-antigen region, or only few faint high-molecular bands observed in *Planktohrix* LPS samples. In addition to distinct migration patterns, significant variations in the overall intensity of ProQ staining were also noted across various LPS samples. This could reflect differences in the chemical composition of the isolated LPS. Variations in staining intensity for LPSs from different cyanobacterial or bacterial strains have been previously reported using both ProQ Emerald and silver staining [37,44,45,46]. The ProQ Emerald stain is thought to interact more strongly with hydrophobic and poorly charged LPSs [44], while silver staining has a lower affinity for detecting acidic sugars [45].

Rough and semi-rough LPS types, which are typically present in LPS isolates from G- bacteria with high endotoxin activities, such as *E. coli*, *P. aeruginosa* [33,34,35,36,37,39], *Bordetella*, or *Salmonella* [46,47], were not detected in most LPS isolated from cyanobacterial biomasses. Despite this, some samples displayed high levels of endotoxin activity (HAB-Mi and *Microcystis* PCC7806 LPS) or pro-inflammatory effects (HAB-Pl, HAB-Do, and HAB-Mi LPSs in intestinal cells), which were comparable to equivalent concentrations of *E. coli* or *P. aeruginosa* LPS. This suggests that LPSs with an *E. coli*-like profile were not the primary drivers of the observed activities and effects in the majority of LPS samples from cyanobacteria in this study. Interestingly, the absence of rough and the prevalence of smooth LPS types were previously reported for LPSs extracted from *M. aeruginosa* NIES-87 [45]. Its electrophoretic profile revealed a smooth-type LPS with a ladder-like pattern with three dominant bands [45], which corresponds well to the migration pattern observed in this study for LPSs of *Microcystis*-dominated blooms HAB-Mi as well as the *Microcystis* PCC7806 culture, both showing LPSs with laddering bands in the smooth LPS region, with distinct major intense bands in the intermediate and very long O-antigen region. Recently, *Geitlerinema* HCC1097 was found to possess a smooth-type-like LPS, characterized by the presence of two bands without a repetitive pattern in the region of the long/very long O-antigen [25]. In our study, only few slowly migrating bands without a laddering pattern were observed in LPSs from HAB-Pl and the *Planktothrix* NIVA-CYA126/8 culture. Despite sample-specific electrophoretic profiles in this study, all LPSs from various cyanobacterial biomasses showed the presence of one or few high-density bands with higher molecular weights, indicating the possible dominance of smooth LPS-type molecules with a certain type of structure (e.g., specific number of repeating sugar units in O-specific antigen chain).

The hypothesis of G- bacterial LPSs having a significant impact on the overall biological activity of the respective sample seems to be supported by the fact that *Dolichospermum*- and *Microcystis*-dominated water bloom LPSs showed the highest biological activity in all used cell models. At the same time, they had the highest proportion of G- bacteria in the biomass (50.3% and 7.4% of analyzed DNA sequences, respectively). On the other hand, the third highest share of the G- bacterial genome was found in the biomass of *Aphanizomenon*-dominated CyanoHAB (3.2%) and its LPS was completely inactive in enterocyte models. The only effect observed was the activation of TNFα production by macrophages in the highest used concentration. In contrast, *Planktothrix*-dominated CyanoHAB had only low levels of G- bacteria (0.5% of analyzed DNA sequences), while its LPS induced significant production of IL-8 by HT-29 cells. The intensity of the response was comparable with the most active CyanoHAB samples. In addition, the response was comparable to *E. coli* and even higher than that of *P. aeruginosa* LPSs, i.e., LPS from G- bacteria. These seemingly contradictory results might be explained by the different bioactivity of the LPS in respective mixtures. It is well known that many of the bacterial LPSs show high bioactivity. The best-known example is LPSs from *E. coli*, a model ligand for TLR4 inducing pro-inflammatory responses via its activation [17]. LPSs from the strain O113:H10 have even become the standard for evaluating endotoxin activity according to the World Health Organization (WHO) [48]. On the other hand, there are also bacterial species with LPSs bearing anti-inflammatory activity, e.g., *Rhodobacter sphaeroides* [49] whose LPS is commercially available as an antagonist for TLR4 (InVivogen). As the microbial community of water blooms is very diverse and dynamic [8,9,10,42,50], various G- bacteria can be present in the studied biomasses and the pro- and anti-inflammatory properties of the LPSs can manifest and support or weaken each other in the environmental mixture. Moreover, different cyanobacterial LPSs can also elicit either pro- or anti-inflammatory effects [2].

The results obtained with LPSs isolated from laboratory cultures, particularly the *Dolichospermum* culture, provide additional evidence that the predominant presence of G- bacteria may not entirely account for the biological activity of certain LPSs. In the biomass, only 1.7% of G- bacteria was detected but the LPS showed, together with *Aphanizomenon* LPS, significant pro-inflammatory effects in all cell types. In the case of Caco-2 cells, it was the only LPS significantly active. These effects were, in most cases, comparable to or even higher than those caused by positive controls, pure G- bacterial LPSs. In contrast, the cyanobacterial culture with the highest share of G- bacteria (10.2%), *Planktothrix*, produced LPSs with almost no biological activity. The only effect of this LPS was an induction of TNFα production by macrophages in the highest used concentration but *Aphanizomenon* and *Dolichospermum* LPS activated the cells in concentrations twenty times or even a hundred times lower. Interestingly, LPSs isolated from the *Microcystis* culture had no biological activity in any of the cell models despite the fact that it showed high endotoxin activity similar to *Aphanizomenon* LPSs and more than a thousand times higher than the most active *Dolichospermum* LPS. This is in contrast to results obtained by Bláhová et al. who observed an induction of TNFα and IL-6 production by three *Microcystis* LPS isolates (1 µg/mL) in whole human blood [1]. On the other hand, significant induction of neutrophils occurred only after using a concentration of LPS one order of magnitude higher (10 µg/mL) [24].

Importantly, induction of pro-inflammatory chemokine IL-8 by differentiated Caco-2 cells, the enterocyte-like cells, was accompanied by an increase in levels of other molecules playing various roles in inflammation. Besides IL-8, other chemokines (GROα, I-TAC, MCP-1, MIP-1α, MIP-1β, RANTES, and SDF-1) were induced. They act as chemoattractants for neutrophils, T-cells, monocytes/macrophages, and/or other types of immune cells. These cells migrate to the site of primary activation and they take a part in the initiation and development of inflammation [51,52,53,54]. Other induced molecules were a wide spectrum of pro-inflammatory cytokines. IL-1α, IL-1β, IL-5, IL-6, IL-17, IL-32α, and TNFα were induced by *Dolichospermum*- and *Microcystis*-dominated CyanoHAB LPSs. These cytokines activate the immune cells to promote and resolve the inflammation [55,56,57]. On the other hand, anti-inflammatory cytokines are also released to keep the inflammation non-destructive [56,58]. Various molecules from this group were increased in our experiments (IL-2, IL-10, IL-12, IL-27, and sTREM-1). Altogether, we can see that the LPSs from CyanoHABs can activate a wide range of bioactive molecules involved in the induction, promotion, and regulation of inflammation *in vitro*.

## 4. Conclusions

Based on our results, we can conclude that complex environmental mixtures of cyanobacterial water bloom LPSs can have pro-inflammatory effects on the intestinal epithelium as well as immune cells *in vitro*. CyanoHABs with different dominant taxa or different cultured cyanobacterial strains contained LPSs with varying structural compositions, significantly different endotoxin activities, or *in vitro* pro-inflammatory effects, in some instances comparable to the activities and effects of purified LPSs from G- bacteria *E. coli* or *P. aeruginosa*. However, we did not observe clear relationships between major dominant cyanobacterial taxa, endotoxin activities, pro-inflammatory *in vitro* effects, electrophoretic profiles of isolated LPS, or the presence of G- bacterial DNA in the cyanobacterial biomass. Further research is needed to comprehend the connection between the specific microbial composition of CyanoHABs, the diversity of structure in LPS derived from both cyanobacteria and G- bacteria associated with blooms, and the structure-dependent differences in their bioactivity. This will help identify the key drivers and factors behind the endotoxin activity and pro-inflammatory effects of the complex mixtures of LPSs from CyanoHABs. It will also allow for accurate assessment and monitoring of exposure to pro-inflammatory components of water bloom biomasses, and thus enable proper risk assessment and predictions of impacts on human health.

## 5. Materials and Methods

### 5.1. Cyanobacterial Biomass Preparation

Cyanobacterial water blooms were collected from water reservoirs in the Czech Republic (Table 1). Cyanobacterial biomass was concentrated using a plankton net (20 µm mesh), transported on ice to the laboratory as described in detail in previous publications [29,30,31,32], frozen, and lyophilized. Species determination and cell counting were performed by microscopic evaluation of samples fixed with 2% formaldehyde. The cell counts were converted to biovolumes according to established databases and methods [59,60] and expressed as a % share of biovolume in a particular sample. According to the dominant species, the biomasses were labeled as follows: cyanobacterial water bloom dominated by *Aphanizomenon klebahnii* as HAB-Ap, *Planktothrix aghardii* as HAB-Pl, *Dolichospermum curvum* as HAB-Do, and *Microcystis aeruginosa* as HAB-Mi (Table 1).

Laboratory cultures of cyanobacteria were obtained from Pasteur Culture Collection of Cyanobacteria (PCC, Paris, France), Culture Collection of Algae of the Norwegian Institute for Water Research (NIVA, Oslo, Norway), and Culture Collection of Autotrophic Organisms (CCALA, Třeboň, Czech Republic): *Aphanizomenon flos-aquae* PCC7905 (Ap), *Planktothrix agardhii* NIVA-CYA126/8 (Pl), *Dolichospermum flos-aquae* CCALA007 (Do), and *Microcystis aeruginosa* PCC7806 (Mi). Cyanobacterial cultures were grown in a mixture of Zehnder medium, Bristol (modified Bold) medium, and distilled water in the ratio 1:1:2 (*v*/*v*/*v*) under continuous illumination (cool white fluorescent tubes, 3000 lux) at 22 °C ± 2 °C [24]. The cultures were handled aseptically and grown in filter-cap tissue culture flasks (TPP, Trasadingen, Switzerland) without active aeration. Cyanobacterial biomass was collected from the cultures approaching the stationary phase of the growth, harvested by centrifugation (2630× *g*/5 min), and freeze-dried. 

### 5.2. LPS Preparation

A fraction of cyanobacterial LPS was obtained by hot phenol–water extraction as reported previously [1,24,41]. Suspension of the lyophilized cyanobacterial biomass (1 g) in MilliQ water (50 mL) was sonicated using an ultrasonic bath (10 min), then heated to 68 °C, mixed with 50 mL of pre-warmed 90% phenol, and stirred for 20 min at 68 °C. After cooling to 4 °C, the mixture was centrifuged (5630× *g*/30 min/4 °C), and the supernatant (aqueous phase with LPS) was collected. The phenol layer was re-extracted with 50 mL of MilliQ water. Pooled supernatants were purified by 48 h of dialysis using cellulose membranes (33 × 21 mm, Sigma–Aldrich, D9652, St. Louis, MI, USA) against MilliQ water (1 L) containing toluene (10 μL/L) to avoid bacterial contamination (water was changed after 24 h). Dialyzed extracts were centrifuged (5630× *g*/30 min/4 °C) and collected supernatants lyophilized. The semi-purified freeze-dried extract of LPS was resuspended in 3.75 mL of 0.1 M Tris–HCl buffer (pH 7.4) containing 25 µg/mL of ribonuclease A (RNase, Sigma–Aldrich, R4642) and incubated for 16 h at 37 °C. Then, 3.75 mL of 90% phenol in 0.1 M Tris-HCl was added, vortexed, and incubated for 4 min at room temperature (RT). The solution was centrifuged (18,410× *g*/15 min/RT), and the aqueous phase was separated and purified for 48 h by dialysis for a second time (see above) and then lyophilized. The freeze-dried powder constituting purified LPS was weighed for assessment of the content of LPSs in the biomass and maintained at −20 °C. Isolated LPSs were dissolved to a concentration of 10 mg/mL either in MilliQ water for characterization by SDS-PAGE, or in PBS with 0.1% BSA (*v*/*v*, Sigma-Aldrich) for PyroGene^TM^ rFC or *in vitro* cell assays.

### 5.3. Characterization of LPSs by SDS-PAGE

Isolated LPSs were separated by SDS-PAGE [61]. LPS samples (10 µg) were diluted with MilliQ water and 4× Laemmli loading buffer (Bio-Rad, Hercules, CA, USA, 161-0747) containing 2-mercaptoethanol (final concentration 357 mM), and heated for 10 min at 65 °C. Then, 4% (*w*/*v*) acrylamide stacking gels were used in combination with 10–20% (*w*/*v*) gradient separating gels (1 mm thickness) prepared from 40% (*w*/*v*) acrylamide/bis acrylamide solution, 37.5:1 (Serva, Heidelberg, Germany) using a Mini-Protean gel casting and electrophoresis system (Bio-Rad). Electrophoresis was conducted at 130 V, and the gels were then washed with deionized water and stacking gels were removed. LPSs were stained with the Pro-Q^TM^ Emerald 300 Lipopolysaccharide Gel Stain Kit (Thermo Scientific, Waltham, MA, USA) according to the manufacturer’s protocol. Washed gels were fixed with 50% (*v*/*v*) methanol-5% (*v*/*v*) acetic acid for 45 min at RT and then washed with 3% (*v*/*v*) acetic acid for 2 × 15 min. LPSs were oxidized for 30 min with a periodic acid-based oxidizing solution supplied with the kit. The gels were washed again with 3% (*v*/*v*) acetic acid (3 × 15 min) and then incubated for 90 min with 1× Pro-Q^TM^ Emerald 300 staining buffer. Prior to UV-light visualization and documentation (Alliance Q9 Advanced, Uvitec Cambridge, UK), the gels were washed with 3% acetic acid for 2 × 20 min.

SDS-PAGE gels with LPS samples were also stained with Coomassie blue to detect eventual protein contaminations. Washed gels were fixed with 40% (*v*/*v*) ethanol-20% (*v*/*v*) acetic acid for at least 1 h at RT, rinsed 5 min with deionized water, and stained overnight with colloidal 0.1% (*w*/*v*) Coomassie Brilliant Blue G250 (Serva) dissolved in 17% (*w*/*v*) ammonium sulfate-34% (*v*/*v*) methanol-3% (*v*/*v*) phosphoric acid. The gels were washed 2 × 5 min with deionized water and de-stained with 10% (*v*/*v*) methanol-10% (*v*/*v*) acetic acid for 3 × 10 min, then with water until the background was sufficiently low. De-stained gels were photo-documented under white light (Alliance Q9 Advanced system). Commercially available LPSs purified from *Escherichia coli* serotype O111:B4 (Sigma-Aldrich, L2630) and *Pseudomonas aeruginosa* serotype 10 (Sigma-Aldrich, L9143) were used as positive controls.

### 5.4. RT-qPCR Detection of Cyanobacteria, Heterotrophic Bacteria, and Gram-Negative Heterotrophic Bacteria

The extraction of DNA from HAB biomass was conducted according to Sehnal et al. [62]. Briefly, 4–10 mg of freeze-dried biomass was resuspended in 0.5 mL of 0.15 M NaCl/0.1 M EDTA solution and homogenized using three freeze–thaw cycles with liquid nitrogen. Then, samples were centrifuged (10 min, at 7200× *g*), the supernatant was discarded, the pellet was resuspended in 0.5 mL of TE buffer, and then 1 μL of RNase (10 mg/mL) was added and the samples were incubated at 37 °C. After 1 h of incubation, 100 μL of lysozyme (50 mg/mL) was added and samples were incubated at 37 °C. After 30 min of incubation, 10 μL (50 mg/mL) of protein kinase K and 2% (final concentration) of sodium dodecyl sulfate (SDS) were added, and samples were incubated at 55 °C for 1 h. Then, selective precipitation was performed by adding 150 μL of 5 M NaCl to the tubes followed by 0.1 volumes (of total volume) of 10% cetyltrimethylammonium bromide (CTAB) stock solution, mixed by inversion, and incubated at 65 °C for 10 min. For purification, 1 volume of chloroform was added, and tubes were incubated on ice for 30 min to allow for protein precipitation. Then, samples were centrifuged (10 min, 7200× *g*, 4 °C), and the supernatant was transferred to fresh tubes, mixed with 0.6 volumes of isopropanol, and incubated overnight at 4 °C. Following this, samples were centrifuged (30 min, 16,000× *g*, 4 °C), the isopropanol supernatant was discarded, and the pellet was washed twice with 1 mL of 70% ethanol. Finally, samples were centrifuged (30 min, 16,000× *g*, 4 °C), the supernatant discarded, and the pellet air-dried and then resuspended in 30 μL of TE buffer. The concentration and quality of the extracted DNA were checked via NanoDrop (Thermo Scientific) and agarose gel electrophoresis. 

qPCR and specific FAM-BHQ1-labeled Taqman probes targeting 16S rRNA genes (Table 2) were used to quantify DNA from (1) cyanobacteria, (2) total heterotrophic bacteria, and (3) a subset of G- heterotrophic bacteria [63]. qPCR was conducted in a LightCycler^®^ 96 Real-time PCR System (Roche, Basel, Switzerland) under the same conditions as in [63]. The results were expressed as a % share of a group-specific DNA (i.e., cyanobacteria, heterotrophic bacteria, G- bacteria) on the total quantified DNA sequences in the sample, i.e., sum of cyanobacteria and total heterotrophic bacteria (=100%). The share of heterotrophic bacteria other than G- (the rest of bacteria) was determined by subtracting the share of G- bacteria DNA from total heterotrophic bacteria DNA.

### 5.5. PyroGene^TM^ rFC Assay

The endotoxin activity of isolated LPS fractions was determined using the PyroGene^TM^ Recombinant Factor C Endpoint Fluorescent Assay (Lonza, Basel, Switzerland). An amount of 100 µL of LPS samples was mixed in 96-well microplate wells with 100 µL of reaction solution (rFC enzyme solution, rFC assay buffer, and fluorogenic substrate in the ratio of 1:4:5) and incubated for 1 h at 37 °C. Fluorescence was measured (excitation 390 nm, emission 440 nm) using an Infinite M200 microplate reader (Tecan, Mannedorf, Switzerland), and endotoxin activity was calculated according to the standard curve. Data were expressed as endotoxin activity in endotoxin units (EU) per mg of dry weight (d. w.) biomass and EU per mg of LPS.

### 5.6. Cell Cultures

All cells were cultivated at 37 °C, in 5% CO_2_/95% air atmosphere. Caco-2 cells (human colorectal adenocarcinoma cell line; CLS, Eppelheim, Germany) were maintained in high-glucose Dulbecco’s modified Eagle’s medium (DMEM; Gibco—Thermo Fisher Scientific, Waltham, MA, USA) supplemented with 10% heat-inactivated low-endotoxin fetal bovine serum (LE FBS; *v*/*v*, PAA), 1% non-essential amino acids (MEM NEAA; *v*/*v*, Gibco), 1 mM sodium pyruvate (Gibco), 2 mM L-glutamine (Gibco), and 100 U/mL of penicillin and 100 µg/mL of streptomycin (Pen/Strep; Gibco). The cells were seeded at a density of 2 × 10^4^ cells/cm^2^ on transparent PET filter inserts (0.4 µm pore size; BD Falcon) in 24-well plates (TPP, Trasadingen, Switzerland) and maintained for 21 days to reach a differentiated monolayer. 

HT-29 cells (human colorectal adenocarcinoma cell line; CLS) were cultivated in high-glucose DMEM supplemented with 10% LE FBS, L-glutamine, sodium pyruvate, and Pen/Strep (Gibco). The cells were seeded at a density of 2 × 10^4^ cells/cm^2^ on 24-well plates (TPP). 

RAW 264.7 cells (mice macrophage cell line; ECACC, Porton Down, UK) were cultured in high-glucose DMEM supplemented with 10% LE FBS, L-glutamine, sodium pyruvate, and Pen/Strep (Gibco). The cells were seeded at a density of 4.5 × 10^4^ cells/cm^2^ on 24-well plates (TPP). 

After differentiation (Caco-2) or 24 h after seeding (HT-29, RAW 264.7), the culture medium was changed and the cells were exposed to the LPS (concentrations are mentioned in graphs) or solvent (0.1% BSA in PBS). Commercially available LPSs from *E. coli* O111:B4 (Sigma-Aldrich, L8274) or *P. aeruginosa* serotype 10 (Sigma-Aldrich, L9143) were used as positive controls for immune responses. As a negative control, untreated cells were used. After 24 h of exposure, the medium was collected for subsequent analyses.

### 5.7. Lactate Dehydrogenase Assay

The potential cytotoxic effect of the LPS was determined by measurement of lactate dehydrogenase (LDH) activity in cell culture medium using the Cytotoxicity Detection KitPLUS (Roche, Pleasanton, CA, USA), as described previously [64]. Briefly, 100 μL of the medium was mixed with the reaction mixture in a ratio of 1:1 and incubated at RT. The absorbance was measured at 492 nm using a SPECTRA Sunrise microplate reader (Tecan, Mannedorf, Switzerland). Untreated cells lysed by supplier-provided lysis buffer were used as a positive control.

### 5.8. Specific Cytokine Detection

The concentration of pro-inflammatory cytokines in the cultivation media was determined by commercially available ELISA kits/assays (Mouse IL-6 Uncoated ELISA Kit; Mouse TNFα Uncoated ELISA Kit; Human IL-8 Uncoated ELISA Kit, Invitrogen—Thermo Fisher Scientific, Waltham, MA, USA). The assays were performed according to the manufacturer’s instructions as described previously [65]. The absorbance was measured using a SPECTRA Sunrise microplate reader (Tecan, Männedorf, Switzerland).

### 5.9. Cytokine Array

Production of a wide range of cytokines by Caco-2 cells was determined using Proteome Profiler Human Cytokine Array Panel A (R&D Systems, Minneapolis, MN, USA) according to the manufacturer’s instructions. Briefly, 1 mL of pooled media from 4–7 independent experiments was mixed with 0.5 mL of Array Buffer and 15 µL of Detection Antibody Cocktail. The reaction mixture was added to previously blocked array membranes for overnight exposure. After the incubation and washing, each membrane was exposed to Streptavidin-HRP solution for 30 min and then activated by Chemi Reagent Mix. Chemiluminescence was captured on X-ray film in the form of dark dots. Afterward, the film was scanned and the optical density of dots was measured by ImageJ software (National Institutes of Health, Bethesda, MD, USA).

### 5.10. NO Production

Changes in NO production were measured indirectly, by virtue of the accumulation of nitrites (the end-product of NO metabolism) into the medium using Griess assay, as described previously [66]. Equal volumes of medium and Griess reagent (50 µL, 40 mg/mL; Sigma-Aldrich) were mixed in a 96-well plate and incubated for 15 min at RT. The absorbance was measured at 540 nm using a SPECTRA Sunrise microplate reader (Tecan). The absorbance of the reaction product was read at 540 nm and the results were calculated according to the sodium nitrite calibration curve.

### 5.11. Statistical Methods

Data are presented as mean ± standard error of the mean (SEM). The number of independent experiments (n) is stated in the figure legends. Statistical analysis was performed using GraphPad Prism version 6.01 for Windows (GraphPad Software, La Jolla, CA, USA). Data were statistically analyzed using One-way ANOVA followed by Dunnett’s multiple comparison test. In the case of non-homogenic data variance, the Kruskal–Wallis test was used. Data converted to control (LDH assay) were analyzed by the one-sample *t*-test. Statistically significant results were marked ** for *p* < 0.01 and * for *p* < 0.05.

## Figures and Tables

**Figure 1 toxins-15-00169-f001:**
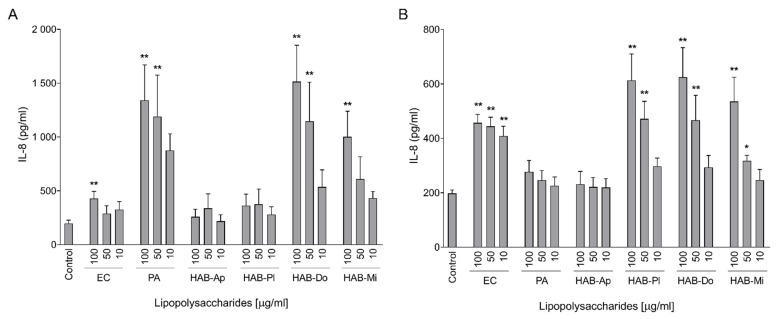
Production of IL-8 by differentiated Caco-2 (**A**) and HT-29 (**B**) cells after exposure to LPSs isolated from water blooms. IL-8 concentrations were measured using ELISA after 24 h of exposure to 10, 50, and 100 μg/mL of studied LPSs and positive controls *E. coli* and *P. aeruginosa*. Effects of LPS treatments were compared with untreated cells (Control). Data are expressed as the mean ± SEM; n ≧ 4. Data were analyzed using ANOVA followed by Dunnett’s multiple comparison test (** *p* < 0.01 and * *p* < 0.05).

**Figure 2 toxins-15-00169-f002:**
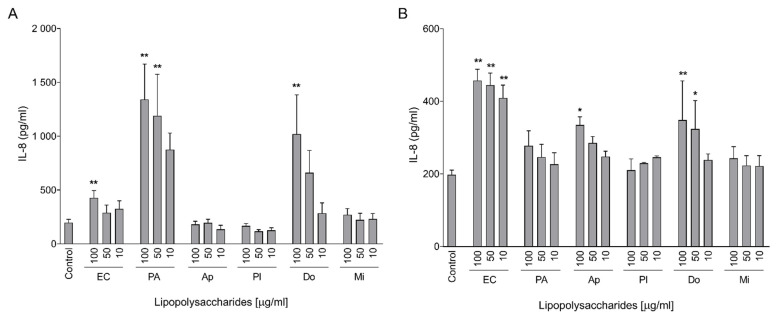
Production of IL-8 by differentiated Caco-2 (**A**) and HT-29 (**B**) cells after exposure to LPSs isolated from cyanobacterial laboratory cultures. IL-8 concentrations were measured using ELISA after 24 h of exposure to 10, 50, and 100 μg/mL of studied LPSs and positive controls *E. coli* and *P. aeruginosa*. Data are expressed as the mean ± SEM; n ≧ 3. Data were statistically analyzed using ANOVA followed by Dunnett’s multiple comparison test (** *p* < 0.01 and * *p* < 0.05).

**Figure 3 toxins-15-00169-f003:**
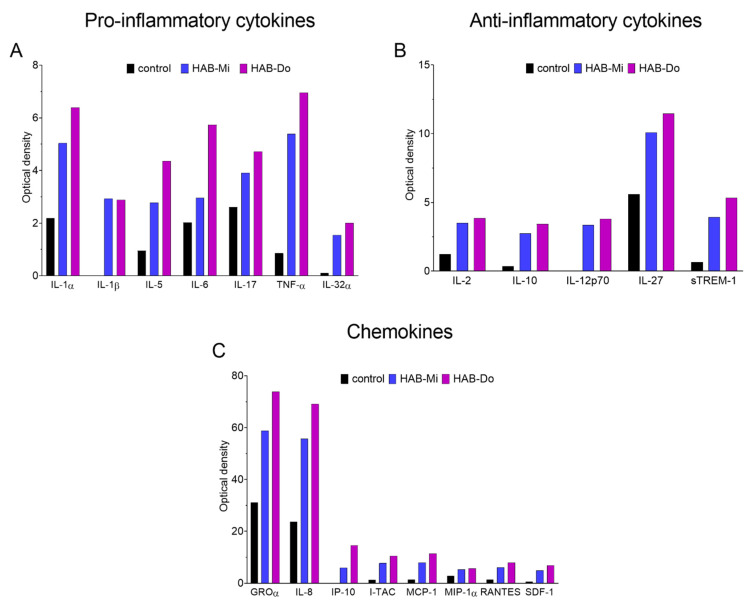
Cytokine array. Medium from 4–7 independent experiments with differentiated untreated Caco-2 cells (control), as well as cells exposed to 100 µg/mL of HAB-Mi and HAB-Do for 24 h, was collected and pooled, and the cytokine array was performed. The optical density of each dot on the membrane was determined, duplicates were averaged, and the obtained values were plotted. Pro-inflammatory (**A**) and anti-inflammatory (**B**) cytokines as well as chemokines (**C**) induced after the exposure to both LPSs were chosen and shown.

**Figure 4 toxins-15-00169-f004:**
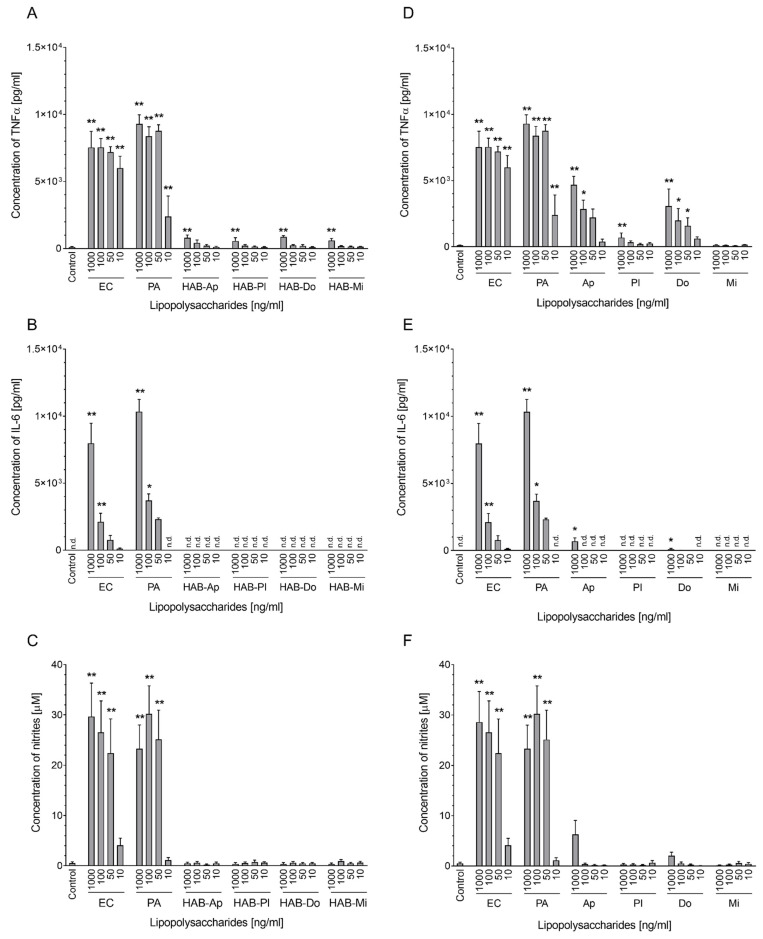
Production of TNFα (**A**,**D**), IL-6 (**B**,**E**), and NO (**C**,**F**) by RAW 264.7 cells after exposure to LPSs isolated from water blooms (**A**–**C**) and laboratory cultures (**D**–**F**). Concentrations of interleukins were measured by ELISA, and concentrations of nitrites as an indirect way to determine NO production were measured using the Griess reaction, both after 24 h of exposure to 10, 50, 100, and 1000 ng/mL of studied LPS and positive controls *E. coli* and *P. aeruginosa* LPSs. Effects of LPS treatments were compared with untreated cells (Control). Data are expressed as the mean ± SEM; n ≧ 3. Data were statistically analyzed using ANOVA followed by Dunnett’s multiple comparison test or Kruskal–Wallis test in the case of non-homogenous data variation (** *p* < 0.01 and * *p* < 0.05). n.d. = not detected.

**Figure 5 toxins-15-00169-f005:**
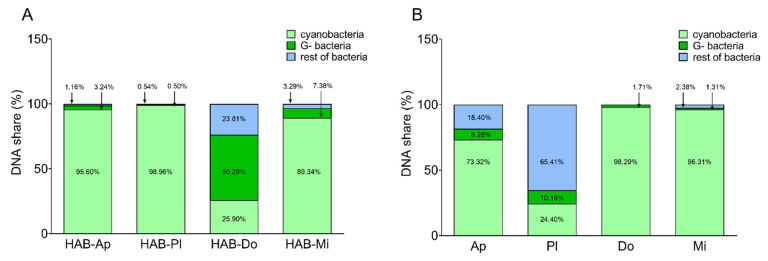
qPCR quantification of DNA sequences in water blooms (**A**) as well as cell culture (**B**) biomasses. TaqMan Probes specific for total heterotrophic bacteria, Gram-negative bacteria (G-), and cyanobacteria were used to quantify their DNA. The results are expressed as a share (%) of cyanobacteria, G-, and the rest of the bacteria (i.e., total heterotrophic bacteria without G-) in total quantified DNA sequences. Water bloom samples (HAB) were dominated by the genera *Aphanizomenon* (Ap), *Planktothrix* (Pl), *Dolichospermum* (Do), and *Microcystis* (Mi). Laboratory cultures represent *Aphanizomenon* PCC7905 (Ap), *Planktothrix* NIVA-CYA 126/8 (Pl), *Dolichospermum* CCALA007 (Do), and *Microcystis* PCC7806 (Mi).

**Figure 6 toxins-15-00169-f006:**
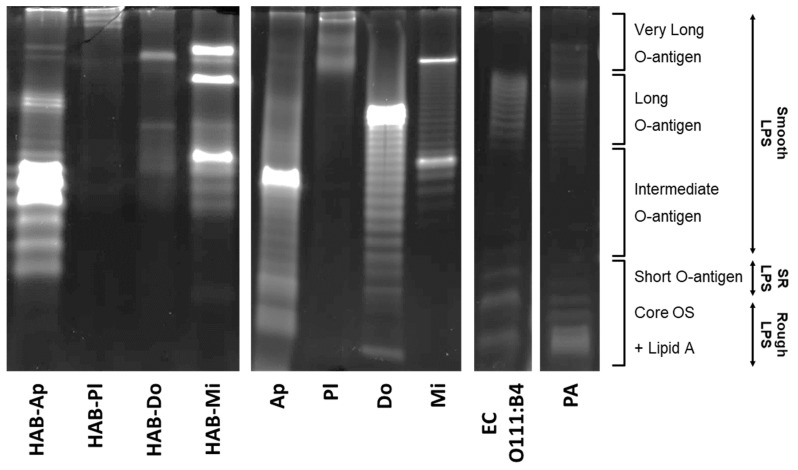
SDS-PAGE of LPS followed by Pro-Q^TM^ Emerald 300 staining. LPSs were isolated from cyanobacterial water blooms (HAB) dominated by *Aphanizomenon* (HAB-Ap), *Planktothrix* (HAB-Pl), *Dolichospermum* (HAB-Do), and *Microcystis* (HAB-Mi), or from cultured cyanobacteria: Ap—*Aphanizomenon* PCC7905, Pl—*Planktothrix* NIVA/CYA 126/8, Do—*Dolichospermum* CCALA007, Mi—*Microcystis* PCC7806. Positive controls: EC—LPS from *Escherichia coli* serotype O111:B4 and PA—LPS from *Pseudomonas aeruginosa*. SR—semi-rough LPS. Core OS—core oligosaccharide. Descriptions of LPS structures and types correspond to the published SDS-PAGE separation patterns of LPS isolated from *Escherichia coli* and *Pseudomonas aeruginosa* [33,34,35,36,37,39]. All LPS samples were loaded at 10 µg per lane.

**Table 1 toxins-15-00169-t001:** Endotoxin activity of LPS isolated from water bloom biomasses as well as cyanobacterial cultures, and positive controls measured by PyroGene^TM^ assay. For CyanoHABs, cyanobacterial compositions of the biomasses are indicated. n.d. = not detected.

Sample Code	Locality	Dominating Species (% Biovolume)	mg LPS/g d.w. (±S.E.)	PyroGene^TM^ (×10^3^ EU/mg LPS)	PyroGene^TM^ (EU/mg d.w.)
**Water blooms**					
HAB-Ap	Ruda 25 July 2013	*Aphanizomenon klebahnii* (92.4%) [29]	15.88 (±4.61)	0.43	6.9
HAB-Pl	Kamenický Šutrák 28 August 2014	*Planktothrix agardhii* (92%) [30]	7.85 (±3.09)	0.17	1.3
HAB-Do	Staňkov 11 September 2013	*Dolichospermum curvum* (73.5%) [31]	12.35 (n.d.)	0.58	7.1
HAB-Mi	Nové Mlýny, 14 August 2012	*Microcystis aeruginosa* (99.8%) [32]	7.13 (±1.16)	37,202. 67	265,092.9
**Cyanobacterial cultures**					
Ap		*Aphanizomenon flos-aquae* PCC 7905	45.98 (n.d.)	1237.34	21,158.44
Pl		*Planktothrix aghardii* NIVA/CYA 126/8	7.60 (n.d.)	2.01	6.96
Do		*Dolichospermum flos-aquae* CCALA 007	23.43 (±1.54)	0.67	14.63
Mi		*Microcystis aeruginosa* PCC 7806	13.24 (±2.89)	1078.50	9870.34
**Gram-negative bacteria (positive controls)**					
EC		*Escherichia coli* O111:B4		3689.95	
PA		*Pseudomonas aeruginosa*		387.66	

**Table 2 toxins-15-00169-t002:** List of specific primers and FAM-BHQ1-labeled probes used for Taqman qPCR (Adapted from reference [63]).

Target	Primer Sets and Probe Sequences (5′->3′)
Cyanobacteria	Forward: ACGGGTGAGTAACRCGTRA
	Reverse: CCATGGCGGAAAATTCCCC
	Probe: CTCAGTCCCAGTGTGGCTGNTC
Total heterotrophic bacteria	Forward: TCCTACGGGAGGCAGCAGT
	Reverse: GGACTACCAGGGTATCTAATCCTGTT
	Probe: CGTATTACCGCGGCTGCTGGCAC-3′
Gram-negative bacteria	Forward: AACTGGAGGAAGGTGGGGAT
	Reverse: AGGAGGTGATCCAACCGCA
	Probe: GACGTAAGGGCCATGAGGACTTGACGTC

## Data Availability

The data are contained within the article and Appendix A.

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
