# Peer review of "Cyanobacterial Harmful Bloom Lipopolysaccharides Induce Pro-Inflammatory Effects in Immune and Intestinal Epithelial Cells In Vitro"

_toxins, 2023, doi:10.3390/toxins15030169_

Round 1
Reviewer 1 Report
The manuscript describes experiments on the effect of cyanobacterial LPS of various origin on human cell lines. The study is very extensive and methods and results are described in detail (with a few exceptions, see PDF). It is a valuable contribution to the field and provides new data.
However, I do not agree with the conclusion that cyanoLPS is a health hazard. To substantiate the claim of a health hazard through cyanoLPS a solid estimate of a realistic exposure should be provided and set in the context of the experimental concentrations. This is, how much cyanobacterial biomass needs to be ingested theoretically to result in LPS concentrations in the intestine’s lumen that are in a range of effective experimental concentrations. Including information on the release of LPS from cyanobacterial cells in the human gastrointestinal tract (there should published data).
Much of the discussion related to a possible health hazard is not more than a chain of associations. This is a problem with most studies on cyanobacterial LPS: there is very little evidence that cyanoLPS are a health hazard under realistic exposure scenarios. There is no doubt that any LPS (or any cell constituent) will cause an effect on cell lines given the concentration is high enough. In this context the lack of cytotoxicity could be discussed in more detail.
If such an estimate would suggest that under normal conditions cyanoLPS is unlikely to cause health impairments, this would be a result, too!
The effects of cyanoLPS observed in cell lines is mostly much lower than observed for the positive control LPS. Yet, why should cyanoLPS pose a health risk when EC-LPS (and other LPS) apparently does not, although the latter is always present in every human gut?
I would have loved to see the effect of LPS from commercial cyanobacterial food supplements. The recommended comsumption of 2 g d.w. per day is a realistic exposure scenario.Compared to this, unintentional ingestion of bloom material will only very rarely reach this level - some 10 g fresh weight is a substantial mass.
Since the study is technically very sound, I would recommend that the authors make their manuscript more concise, firstly, by removing all redundant parts and, secondly, by leaving out all speculations on health hazards that are not directly substantiated by the results.
Technical issues: I recommend to reorganise the figures in a way that they can be arranged in a way that allows a direct comparison, e.g., figures 3 and 6 in two columns on a single page or figures 4 and 7 scaled smaller and combined in a single figure. The same applies to the tables.

Author Response
We would like to thank the Reviewer for the thoughtful comments which improved the quality of the manuscript. Below are the responses to the main questions, detailed reactions to comments provided in the pdf file are in the attached special version of the revised manuscript to keep the answers clear and easy to connect to the text.
The manuscript describes experiments on the effect of cyanobacterial LPS of various origin on human cell lines. The study is very extensive and methods and results are described in detail (with a few exceptions, see PDF). It is a valuable contribution to the field and provides new data.
However, I do not agree with the conclusion that cyanoLPS is a health hazard. To substantiate the claim of a health hazard through cyanoLPS a solid estimate of a realistic exposure should be provided and set in the context of the experimental concentrations. This is, how much cyanobacterial biomass needs to be ingested theoretically to result in LPS concentrations in the intestine’s lumen that are in a range of effective experimental concentrations. Including information on the release of LPS from cyanobacterial cells in the human gastrointestinal tract (there should published data).
Reply: Thank you for the comment. The statements about health hazard were change according your detailed comments in the pdf.
Much of the discussion related to a possible health hazard is not more than a chain of associations. This is a problem with most studies on cyanobacterial LPS: there is very little evidence that cyanoLPS are a health hazard under realistic exposure scenarios. There is no doubt that any LPS (or any cell constituent) will cause an effect on cell lines given the concentration is high enough. In this context the lack of cytotoxicity could be discussed in more detail.
If such an estimate would suggest that under normal conditions cyanoLPS is unlikely to cause health impairments, this would be a result, too!
Reply: In this study, we focused on immunomodulatory properties of LPS. We assumed that LPS as a part of the environmental mixture could contribute to pro-inflammatory activation of target cells (intestinal epithelium). Such activation could possibly lead to increased permeability of the intestinal barrier and take a part in local inflammation induction. Therefore, only non-cytotoxic concentrations of the LPS were chosen.
The discussion related to possible health hazard has been edited according to the specific comments in the pdf.
The effects of cyanoLPS observed in cell lines is mostly much lower than observed for the positive control LPS. Yet, why should cyanoLPS pose a health risk when EC-LPS (and other LPS) apparently does not, although the latter is always present in every human gut?
Replay: Effect of LPS is dependent on its structure. Therefore, LPS with high bioactivity exist as well as the with low or no bioactivity. The structure of the LPS differs not only among genera or species, it differs also among strains. The LPS used is isolated from pathogenic E. coli strain O111:B4 with LPS showing high immunomodulatory potencies. This specific LPS in not present routinely in guts. Similarly, Pseudomonas aeruginosa is considered as a pathogen and its LPS as the major virulence factor. We chose these LPS as positive controls to see how intensive reaction of our in vitro systems these pathogen-derived compounds would induce and to compare the effects of studied LPS isolated from water bloom biomasses and cyanobacterial cultures.
In the case of enterocyte-like models, at least Dolichospermum-dominated water bloom LPS showed significant activity comparable with or even higher than some of the positive LPS. In differentiated Caco-2 cell model (routinely used for pharmacological studies) the effect was much stronger then that of EC LPS and similar to PA LPS. In HT-29, where PA LPS was ineffective, Dolichospermum- and Aphanizomenon-derived LPS showed significant effects. We see that as an indication that environmental mixtures of LPS might have immunomodulatory potency. And that it deserves other attention in the terms of searching which genera of LPS producers could provide these effects and if sufficient exposure is possible.
I would have loved to see the effect of LPS from commercial cyanobacterial food supplements. The recommended comsumption of 2 g d.w. per day is a realistic exposure scenario.Compared to this, unintentional ingestion of bloom material will only very rarely reach this level - some 10 g fresh weight is a substantial mass.
Replay: We agree, it would be interesting.
Since the study is technically very sound, I would recommend that the authors make their manuscript more concise, firstly, by removing all redundant parts and, secondly, by leaving out all speculations on health hazards that are not directly substantiated by the results.
Replay: We try our best to edit the text according to your recommendations in pdf.
Technical issues: I recommend to reorganise the figures in a way that they can be arranged in a way that allows a direct comparison, e.g., figures 3 and 6 in two columns on a single page or figures 4 and 7 scaled smaller and combined in a single figure. The same applies to the tables.
Replay: The tables as well as mentioned figures were merged as was recommended.

Reviewer 2 Report
This study evaluated LPS content and activity of cyanobacterial blooms and compared to LPS content and activity of two gram negative bacteria and laboratory cultures of the predominant cyanobacterial species in each bloom sample: Aphanizomenon, Dolichospermum, Planktothrix and Microcystis. This information is relevant due to the most common human health effects from contact with blooms are thought to be significantly linked to the LPS content of the bloom although the evidence is lacking.
I have some minor suggestions for the authors.
1) Line 47- removed "they"
2) Line 130- remove "used"
3) Section 2.2.3, Line 210- recommend adding Aphanizomenon with Dolichospermum here and remove the last sentence (lines 213-214) since the NO increase was not significant.
4) Figures 4 and 7- consider adding percentages in graph or in the figure description. It helps the reader.
5) Line 285- I don't believe "the toxic effects of many cyanotoxins are studied deeply". EPA and WHO only have provisional guidance for a few.
6) Line 323- a period missing between the sentences
7) Lines 328-331- the authors make the point that the LPS migration patterns differed between the HAB and the cultures ("major structural differences"), but they appear to have similarities except for the Dolichospermum. In lines 342-345, the authors then discuss the similarities of the Microcystis bloom and culture. Please clarify your statements that seem to contradict.
8) Line 373- The authors state that the Dolichospermum "showed the highest pro-inflammatory effects of all samples in all cell types", but in the RAW cell for IL-6, the Aphanizomenon appears to be higher
Author Response
We would like to thank the Reviewer for the thoughtful comments which improved the quality of the manuscript. Below are the responses to the questions.
This study evaluated LPS content and activity of cyanobacterial blooms and compared to LPS content and activity of two gram negative bacteria and laboratory cultures of the predominant cyanobacterial species in each bloom sample: Aphanizomenon, Dolichospermum, Planktothrix and Microcystis. This information is relevant due to the most common human health effects from contact with blooms are thought to be significantly linked to the LPS content of the bloom although the evidence is lacking.
I have some minor suggestions for the authors.
- Line 47- removed "they"
Reply: The word was deleted.
- Line 130- remove "used"
Reply: The word was deleted (now Line 157).
- Section 2.2.3, Line 210- recommend adding Aphanizomenon with Dolichospermum here and remove the last sentence (lines 213-214) since the NO increase was not significant.
Reply: Aphanizomenon was added (now Line 184)and the last sentence was deleted.
- Figures 4 and 7- consider adding percentages in graph or in the figure description. It helps the reader.
Reply: The percentages were added in the graphs and the figures were merged in the Figure 5.
- Line 285- I don't believe "the toxic effects of many cyanotoxins are studied deeply". EPA and WHO only have provisional guidance for a few.
Reply: According also to comments of the second reviewer, this part of the text was deleted.
- Line 323- a period missing between the sentences
Reply: The missing dot was put at the end of the sentence.
- Lines 328-331- the authors make the point that the LPS migration patterns differed between the HAB and the cultures ("major structural differences"), but they appear to have similarities except for the Dolichospermum. In lines 342-345, the authors then discuss the similarities of the Microcystis bloom and culture. Please clarify your statements that seem to contradict.
Reply: We corrected the text to acknowledge similarities between CyanoHAB LPS and LPS isolated from cultures of corresponding dominating species (except Dolichospermum).
- Line 373- The authors state that the Dolichospermum "showed the highest pro-inflammatory effects of all samples in all cell types", but in the RAW cell for IL-6, the Aphanizomenon appears to be higher.
Reply: Thank you for your comment. The text was rewritten to “In the biomass, only 1.7% of G- bacteria was detected but the LPS showed, together with Aphanizomenon LPS, significant pro-inflammatory effects in all cell types. In the case of Caco-2 cells it was the only LPS significantly active.”
Round 2
Reviewer 2 Report
I was happy with the changes the authors have made. There are no further changes except for a typo on line 205: add "the" so the sentence reads "In the case of cell cultures..."